Colonization and usage of an artificial urban wetland complex by freshwater turtles

Dupuis-Desormeaux Marc 1 marcd2@me.com
Davy Christina 2
Lathrop Amy 3
Followes Emma 4
Ramesbottom Andrew 4
Chreston Andrea 4
http://orcid.org/0000-0001-8152-6992 MacDonald Suzanne E. 5
1 Department of Biology, York University , Toronto, ON , Canada
2 Wildlife Research and Monitoring Section, Ontario Ministry of Natural Resources and Forestry , Peterborough, ON , Canada
3 Department of Natural History, Royal Ontario Museum , Toronto, ON , Canada
4 Toronto and Region Conservation Authority , Toronto, ON , Canada
5 Department of Psychology, York University , Toronto, ON , Canada
Martinelli Luiz
Electronic publication date: 2018 Aug 8
Publication date: 2018
Volume: 6
Electronic Location ID: e5423
Received 2018 Jan 23; Accepted 2018 Jul 17
Copyright: © 2018 Dupuis-Desormeaux et al.
Copyright year: 2018
Copyright holder: Dupuis-Desormeaux et al.
License: This is an open access article distributed under the terms of the Creative Commons Attribution License, which permits unrestricted use, distribution, reproduction and adaptation in any medium and for any purpose provided that it is properly attributed. For attribution, the original author(s), title, publication source (PeerJ) and either DOI or URL of the article must be cited.
License URL: https://creativecommons.org/licenses/by/4.0/

Keywords: Snapping turtle, Midland Painted turtle, Artificial wetland, Lake Ontario, Sex ratio, Common carp, Restoration ecology, Blanding’s turtle, VHF, Turtle road mortality

Funding: Toronto and Region Remedial Action Plan Toronto and Region Conservation Authority This study was partially funded by grants from the Toronto and Region Remedial Action Plan, and the balance of funding was provided by the Toronto and Region Conservation Authority. There was no additional external funding received for this study. The TRCA was involved in the study design.

==============================
Conservation authorities invest heavily in the restoration and/or creation of wetlands to counteract the destruction of habitat caused by urbanization. Monitoring the colonization of these new wetlands is critical to an adaptive management process. We conducted a turtle mark-recapture survey in a 250 ha artificially created wetland complex in a large North American city (Toronto, Ontario). We found that two of Ontario’s eight native turtle species (Snapping turtle (SN), Chelydra serpentina, and Midland Painted (MP) turtle, Chrysemys picta marginata) were abundant and both were confirmed nesting. The Blanding’s turtle (Emydoidea blandingii) was present but not well established. Species richness and turtle density were not equally distributed throughout the wetland complex. We noted SN almost exclusively populated one water body, while other areas of the wetland had a varying representation of both species. The sex ratios of both SN and MP turtles were 1:1. We tracked the movement of Snapping and Blanding’s turtles and found that most turtles explored at least two water bodies in the park, that females explored more water bodies than males, and that 95% of turtles showed fidelity to individual overwintering wetlands. We performed DNA analysis of two Blanding’s turtles found in the created wetlands and could not assign these turtles to any known profiled populations. The genetic data suggest that the turtles probably belong to a remnant local population. We discuss the implications of our results for connectivity of artificial wetlands and the importance of the whole wetland complex to this turtle assemblage.

Introduction

Turtles play an important role in wetlands as omnivores, scavengers, predators, food sources, and nutrient recyclers, and some turtles are considered keystone species (Paine, 1995). Turtle biomass can be very high in comparison to other vertebrates, measuring up to 877 kg/ha in one temperate wetland (Congdon, Greene & Gibbons, 1986). Turtle populations are in decline in North America due to the loss and degradation of habitat, invasive species, pollution, disease, climate change, and overexploitation (Ernst & Lovich, 2009). Road mortality is also a significant threat to several species (Beaudry, DeMaynadier & Hunter, 2008) and is linked to reduced gene flow and male-skewed sex ratios (Gibbs & Steen, 2005).

Ontario has eight native species of turtles, and seven have been documented in Lake Ontario or its tributaries: Midland Painted (MP; Chrysemys picta marginata), Common Snapping (SN; Chelydra serpentina), Northern Map (Graptemys geographica), Spiny Softshell (Apalone spinifera spinifera), Eastern Musk (Sternotherus odoratus), Spotted (Clemmys guttata), and Blanding’s (Emydoidea blandingii). A 7-year synoptic trapping study concluded in 2010 at 16 sites along the Lake Ontario’s shore found only four native turtle species present (MP, SN, Snapping, Blanding’s, and Map). Low water quality and lack of submergent vegetation was likely to blame for the absence of Eastern Musk (DeCatanzaro & Chow-Fraser, 2010). The Spotted turtle is mostly found on the northern shore of Lake Erie, in the Georgian Bay and in a few scattered locations in southwest Ontario. Habitat loss restricts the Ontario distribution of the Spiny Softshell to a few disjunct populations located in south western Ontario.

Habitat loss due to urbanisation disproportionately affects wetland habitat, and is therefore critical to replace (Dudgeon et al., 2006; Mitsch & Gosselink, 2000). Wetlands provide important ecosystem services and are critical to preserve and restore (Bolund & Hunhammar, 1999; Woodward & Wui, 2001; Zedler & Kercher, 2005b). Wetlands represent approximately 14% of Canada’s land area, but development has reclaimed, contaminated, or degraded their ecological functions in urban centres (Price & Waddington, 2000).

Wetlands provide valuable ecosystem services such as regulating water quantity, and minimizing the impact of floods and storm water (Woodward & Wui, 2001; Zedler & Kercher, 2005a). Thus, the conservation, mitigation, and/or restoration of wetlands is a major priority for the Government of Ontario and Ontario conservation authorities (Ontario Ministry of Natural Resources and Forestry, 2017). Restoration ecology has focused a great deal on restoring and creating wetlands in urban areas to compensate for the rapid loss of wetland habitat due to urbanisation, yet restored wetlands consistently underperform reference wetlands on many ecological indicators (Maron et al., 2012; Suding, 2011; Zedler & Callaway, 1999). Restored wetlands can take >100 years to return to reference levels after restoration, or may become permanently altered by novel environmental conditions and never return to a reference state, especially in colder climates (Moreno-Mateos et al., 2012). Colonization of new wetlands by wildlife is dependent on the quality of the habitat, its age, size and its connectivity to source populations (Cosentino, Schooley & Phillips, 2010; Kadoya, Suda & Washitani, 2004; Ruhi et al., 2009, 2012). As a major urban wetland conservation organization, Toronto and Region Conservation Authority (TRCA) has dedicated a large proportion of its annual C$40 million budget to monitoring, protecting, and restoring wetlands in Canada’s largest urban centre. The aim of this study was to investigate if, and to what extent, turtles had colonized these artificial wetlands. For any Blanding’s turtles, we wanted to determine from which wild population they were most likely derived. Our secondary goal was to test if turtles that had recolonized restored wetlands were evenly distributed within the complex, and if species richness and distribution was similar to what could be expected in a natural Lake Ontario coastal wetland. We hope that monitoring and documenting the extent of any colonization will serve to gauge the success of these particular wetland creation efforts and inform other wetland creation or restoration projects.

Methods

Study site

Tommy Thompson Park (TTP)—Toronto, Ontario (43°37′4″N 79°20′33″W)

The City of Toronto, Canada’s largest metropolitan area, is built on a series of watersheds that cut along a north–south axis ending in Lake Ontario. Toronto is historically rich in wetland habitat that has been gradually lost due to urbanisation. In the 1960s, the Toronto Port Authority built a five km long breakwater (the Leslie Street Spit) into Lake Ontario to create additional lands for port infrastructure in conjunction with the opening of the St. Lawrence Seaway. As the Spit grew in size, native and non-native plants began to colonize the land, and it became an important stopover habitat for migrating birds. By 1991, large colonies of cormorants, gulls, and herons nested at the Spit.

Toronto and Region Conservation Authority was tasked to develop and implement a master plan for a public park on the Spit, leading to the creation of TTP. One component has been natural area enhancement, and TRCA has focused efforts on creating a varied wetland complex, to improve aquatic habitat including features such as aquatic vegetation and structural habitat (logs, root balls, stones, gravel beds, sandy beaches, etc.). TRCA continues to tweak wetland design at TTP using an iterative adaptive management process that relies on active monitoring.

Description and history of the individual water bodies at TTP

At TTP, there are three types of created water bodies forming the wetland complex: ponds, embayments, and cells (Fig. 1). Ponds are small and unconnected stand-alone bodies of water. There are two main ponds in the wetland complex, Triangle and Goldfish ponds. Goldfish pond is the smallest pond and was accidentally created through asphalt disposal in the late 1970s. It measures approximately 500 m2 (0.05 ha). TRCA has not undertaken any restoration projects in Goldfish pond and it has been allowed to naturalize without any intervention. Triangle pond was constructed between 1975 and 1978 as a trial confined disposal facility and it covers 0.8 ha. TRCA capped the facility in 1997 and subsequently created a wetland by adding structural habitat (large stones, rocky shoals, woody materials), as well as aquatic and riparian vegetation in 1998.

Figure 1 Map of Tommy Thompson Park, Toronto, Ontario, Canada.

TRCA image 2017.

The TTP embayments were created as coastal wetlands. Embayment A covers 6.25 ha and has benefited from two restoration projects; (a) 2004–2005 to create spawning shoals on the east side of the embayment; (b) 2009 to create backwater refuge and coastal wetland along the west and south shorelines. Embayment B covers 19 ha and has received two restoration projects; (a) 1995–1997 to create northern pike (Esox lucius) spawning channels in the sheltered backwater area to the east of the embayment, as well as structural habitat in the main embayment; (b) 2011 to improve and enhance the coastal wetland conditions along the shoreline of the main embayment. Embayment C covers 24 ha and has seen three restoration projects; (a) 1996–1998 to create structural, wetland, and riparian habitat on the west portion of the embayment; (b) 1998 to create pike spawning channels on the southeast shoreline; (c) 2010 to improve coastal wetland habitat along the east shoreline along with additional structural habitat enhancements throughout the embayment. Embayment D covers seven ha. In 2012–2014 a coastal wetland restoration project isolated the embayment from the lake through the construction of a berm and the installation of a fish and water level control structure. Embayments A, B, and C are open to the lake, with backwaters and fish spawning channels, while Embayment D is connected to the lake via a fish gate with seven cm vertical grates, impeding the travel of large invasive common carp (Cyprinus carpio) into the wetlands (French, Wilcox & Nichols, 1999).

Cells were created as confined disposal facilities to store dredged material from the lower Don River and Keating Channel and eventually capped and transformed into wetlands. Cells are also connected to Lake Ontario via fish gates. Cell-1 was filled to capacity in 1985 and capped. TRCA created 11 ha of terrestrial and aquatic habitat in Cell-1, including a 7.7 ha coastal hemi-marsh between 2003 and 2006. A fish and water level control structure was installed in 2012. Cell-2 covers 10 ha and was filled to capacity in 1997. Cell-2 was capped and a coastal hemi-marsh wetland was created in 2015–2017 by adding structural habitat and vegetation. TRCA expected to install fish and water level control structures in late 2017–2018 and to plant aquatic vegetation into 2018. Cell-3 measures 32 ha and is still accepting dredged material. TRCA does not expect to transform it into a wetland for some time, and hence this cell is currently devoid of emergent vegetation or structural habitat suitable for turtles.

Survey methods

We studied the turtle population using three main methods: a capture-mark-recapture (CMR) methodology, very high frequency (VHF) tracking, and shore-based visual basking surveys.

Capture-mark-recapture

We conducted a CMR study in 2016 and 2017, trapping turtles using hoop nets, basking traps, or seine nets. CMR (Lecren, 1965) is a robust method to determine population densities and trends (Pollock et al., 1990). We used one metre diameter three-ring hoop nets (no.15 net with 6.25 cm mesh from Champlin Net Company, Jonesville, LA, USA), basking traps (manufactured in-house, varying in size from 1 to 2 m2, all with 6.25 cm mesh size) and a 30 m seine net (manufactured in-house, with 6.25 cm mesh). The size of the mesh precluded capturing very small turtles or juveniles, so our demographic information is skewed toward adults. Juveniles were captured opportunistically by hand. We baited the hoop traps with a variety of food, including frozen fish, canned cat food, canned sardines, and frozen chicken, depending on availability. Bait was placed in perforated bags, hung near the back of the traps at such a height that the bait was in the water. The trap was three-quarters submerged in water, in order to always have air available to trapped animals. Captured turtles were removed from the traps and processed on shore, where we recorded weight with a digital scale, body morphology with digital vernier calipers (plastron length, straight carapace length, carapace width, body depth, precloacal length, foreclaw length) and sex (if possible). For MP turtles, we identified males based on the elongated foreclaws and long tail, with the cloaca situated well beyond the limit of the carapace, compared to shorter foreclaws and tails in females. For SN, we used the relative length of the posterior lobe vs. the base—cloaca tail measurement, and by gentle rocking to encourage penile eversion. For Blanding’s turtles, males have a concave plastron and longer tail. We notched the marginal scutes of the shells to give each turtle a unique identifying number (Cagle, 1939). After processing, we returned turtles to the general area of the capture trap. We trapped for 25 days during the months of June and July 2016 and for 24 days in May and June 2017, using up to 20 hoop traps (13 traps in 2016 and 20 traps in 2017), four basking traps and one seine net (2016 only). The seine net was only used 1 day in 2016 (one SN captured) and was unavailable in 2017 and the basking traps had little success in 2016 and were not used in 2017.

We trapped in each water body for a 1–2-week period, depending on water body size and success of trapping. Using a CMR protocol in successive years also allowed us to observe turtles moving from one water body to another. As part of our invasive species protocol, we removed all captured Red-Eared Sliders (Trachemys scripta elegans) and transferred them to an adoption center (Little RES Q, Pefferlaw, ON, Canada).

We used previously recorded locations of visually surveyed turtles to decide in which water body we should trap. Staff and volunteers had reported many turtle sightings in both ponds, in all embayments and in Cells 1 and 2. Although Cell-2 had previously recorded turtle activity, it was under further restoration during the time of this study and was unsuitable for trapping. Consequently, we trapped in seven distinct water bodies (see Table 1): Goldfish pond (2016/2017), Triangle pond (2016/2017), Cell-1 (2016/2017), Embayments A (2016), B (2016), C (2017), and D (2016/2017). Wetlands that produced very few turtles in 2016 were not re-sampled in 2017.

Table 1 Created water bodies at Tommy Thompson Park (TTP), describing type, size, and year of completion.

Name	Type	Size (ha)	Year completed	
Goldfish	Pond	0.05	1970s	
Triangle	Pond	0.8	1998	
Emb-A	Embayment	6.25	2009	
Emb-B	Embayment	19	2011	
Emb-C	Embayment	24	2010	
Emb-D	Embayment	7	2014	
Cell-1	Cell	6	2012	

Statistics

Population estimates: We applied the Chapman correction for small sample sizes to a simple two-sample Lincoln–Petersen index to estimate population size (Chapman, 1951). We estimated the population size for the two main turtle species at TTP where N = [(K+1)(n+1)]/(k+1); N is the estimated population, n is the individuals captured in 2016, K is the captured population in 2017, and k is the population recaptured in 2017 that was marked from a previous capture.

Turtle sizes: Straight-line carapace length is reported using the mean ± 1 standard deviation, with ranges reported as minimum–maximum.

Species representation: We tested for uniformity of individual species distribution through the various water bodies using a Chi-square test.

Sex ratio: We performed a binomial test (non-parametric two-sided) to determine if the sex ratio of the two main turtle species was unbiased. The resulting probability, p, is two-tailed and is significant at 95% confidence level if the resulting p is smaller than 0.025 or greater than 0.975.

VHF tracking

We affixed a 12 g VHF transmitter (model R185; Advanced Telemetry Systems, Isanti, MN, USA) to certain individuals of turtle species considered endangered or species at risk in Ontario (Blanding’s and SN). Using a water epoxy, we glued the transmitters to the back of the carapace, generally offset to one side, with the antenna trailing. We only attached transmitters to turtles >300 g (transmitter weight <5% turtle body weight). In 2016, we aimed to get an even number of males and females and we able to track eight male and eight female SN and three female Blanding’s turtles. After an initial data investigation, we noticed that the female SN were travelling greater distances than the males. Therefore, in 2017 we became more interested in the movement of females, and consequently added new transmitters only to new female SN. We also removed the old VHF transmitters (most of which had depleted their battery power) from any recaptured male SN. As a result, in 2017, we replaced the transmitters on all three female Blanding’s turtles, on four recaptured female SN, added five new female SN to the study, and removed old, non-functional transmitters from three recaptured male SN. We continued to track the remaining five male SN until we lost their VHF signals.

We tracked each individual turtle to a specific water body on a weekly basis during the active period and once monthly over the winter. Hibernacula locations were determined in the fall by locating turtles that had buried themselves under vegetation, logs, stumps or in the mud and reconfirming these locations in the winter. The weekly time period will necessarily underestimate the movement of turtles throughout the wetland complex but we used it as a baseline indication of the minimum habitat use and of exploratory movement at the study site.

Visual surveys

Staff and volunteers at the park also conducted regular visual surveys (using binoculars) of basking turtles from shore and reported results back to the authors. Visual surveys offer a continuous data set starting in 2004. Staff and volunteers also reported opportunistic observations of turtle nests throughout the 2016–2017 study period.

DNA analysis

We collected blood samples from the Blanding’s turtle by caudal venipuncture and stored these samples on FTA cards (Whatman Inc., Little Chalfont, UK), for later DNA analysis to understand the likely source of these individuals, which could have re-colonized TTP from a nearby population or introduced to the site by humans after collection at a distant source site. DNA was extracted using standard methods for nucleated blood (Smith & Burgoyne, 2004) and genotyped using 11 microsatellite loci. These included loci GmuB08, GmuD16, GmuD21, GmuD28, GmuD55, GmuD87, GmuD88, GmuD93, GmuD107, and GmuD121 (King & Julian, 2004), and loci Eb17 and Eb19 (Osentoski et al., 2002). Polymerase chain reaction followed the recipe and thermocycling parameters detailed in Davy, Bernardo & Murphy (2014). We ran the TPP samples alongside a positive control DNA from a Blanding’s turtle genotyped previously in Davy, Bernardo & Murphy (2014) to ensure that the microsatellite genotypes obtained were accurate and comparable to existing genotype data. The new genotypes were compared to an existing database for Ontario Blanding’s turtles (Davy, Bernardo & Murphy, 2014) and used Geneclass2 (Piry et al., 2004) to assign or exclude populations of origin for the new samples. We used Bayesian methods of assignment (Rannala & Mountain, 1997), and Monte–Carlo re-sampling simulation algorithm (Paetkau et al., 2004), with 1,000 simulated individuals and α = 0.05, and then ran the analysis a second time using frequency-based assignment (Paetkau et al., 2004).

Permits

This study was conducted with the approval of York University’s Animal Care Committee (YUACC#2016-16W- and #2017-16W-R1) and under the Ontario Ministry of Natural Resources (Wildlife Scientific Collector’s Authorization numbers # 1083601 and 1085922) and Endangered Species Act Permit for Species Protection or Recovery (AU-B-007-16 and AU-B-006-17).

Results

Trapping surveys

We captured 126 turtles (97 new and 29 recaptured) over 496 trap-days (catch-per-unit-effort, CPUE, 25.4%) in 2016 and 103 turtles (50 new and 53 recaptured) over 514 trap-days (CPUE 20.0%) in 2017 (Table 2). Three of the four expected native turtle species were represented in our trapping: MP, SN, Blanding’s, and the introduced Red-Eared Slider (see Table 2). Based on the trapping results we estimated the number of individual MP turtles within TTP at 140 turtles ± 11 (n = 69, K = 71, k = 35) and the number of SN at 35 turtles ± 3 (n = 20, K = 24, k = 14). Neither MP turtles (df = 6, n = 105) = 143.067, p < 0.001) or SN (df = 7, n = 30) = 23.867, p < 0.001) were uniformly distributed across the individual capture locations. The proportion of MP to SN in the whole wetland complex was 3.5:1 but varied by individual wetlands from 1:12 to 28:1. The male:female sex ratios of both MP (45:44, p = 0.584) and SN (12:15, p = 0.351) did not statistically differ from the expected 1:1 ratio if the demographics of each species are taken as a whole over the entire TTP wetland complex. However, MP turtle sex ratios deviated from the expected 1:1 in two wetlands, where in Triangle pond was male skewed, but just below significance (1.7:1, p = 0.970) and Embayment D was significantly female skewed (1:3.5, p = 0.015).

Table 2 Total number of new turtles captured in each water body surveyed at Tommy Thompson Park, 2016–2017.

Species vs. specific wetland	Midland Painted (Chrysemys picta marginata)	Snapping (Chelydra serpentina)	Blanding’s (Emydoidea blandingii)	Red-Eared Slider (Trachemys scripta elegans)	Total number of individual turtles	Ratio of MP:SN	
Goldfish pond	28 (14, 13, 1)	1 (1, 0, 0)	3 (0, 3, 0)	1 (1, 0, 0)	33	28:1	
Triangle pond	49 (26, 15, 8)	5 (3, 2, 0)	0	1 (0, 1, 0)	55	9.8:1	
Cell-1	1 (0, 1, 0)	12 (4, 6, 1)	0	4 (2, 2, 0)	17	1:12	
Emb-A	1 (1, 0, 0)	1 (0, 0, 1)	0	0	2	1:1	
Emb-B	1 (0, 1, 0)	1 (1, 0, 0)	0	0	2	1:1	
Emb-C	0	1 (0, 1, 0)	0	0	1	0:1	
Emb-D	24 (4, 14, 6)	6 (3, 3, 0)	0	3 (2, 1, 0)	33	4:1	
On land	1 (0, 0, 1)	3 (0, 3, 1)	0	0	4	1:3	
Total	105 (45, 44, 16)	30 (12, 15, 3)	3 (0, 3, 0)	9 (5, 4, 0)	147	3.5:1	
Note:

Each entry represents: total (male, female, unsexed juveniles) and ratio of Midland Painted turtle to Snapping turtle (MP:SN).

We report MP and SN size and mass for the wetland complex and by specific wetland in Table 3. We also captured three mature female Blanding’s turtles (straight carapace length 215, 241, and 230 mm, and mass of 1,356, 2,200, 1,885 g, respectively). Blood samples were successfully collected from two of these females.

Table 3 Sex, mean straight carapace length (SCL) and mass of adult Midland Painted Turtle, Chrysemys picta marginata, and Common Snapping Turtle, Chelydra serpentina, by wetland at the Tommy Thompson Park.

Wetland	Painted	Snapping	
Males	Females	Males	Females	
TTP–Total (n)	(45)	(44)	(12)	(15)	
 SCL (mm)	127 ± 12	146 ± 19	295 ± 66	245 ± 53	
 Mass (g)	263 ± 63	447 ± 148	7,265 ± 4,252	4,108 ± 1,994	
Goldfish Pond- (n)	(14)	(13)	(1)		
 SCL (mm)	125 ± 12	149 ± 9	270		
 Mass (g)	262 ± 59	482 ± 80	3,963		
Triangle Pond- (n)	(26)	(15)	(3)	(2)	
 SCL (mm)	125 ± 11	139 ± 21	368 ± 33	292 ± 4	
 Mass (g)	250 ± 57	401 ± 162	11,833 ± 3,085	5,025 ± 671	
Cell-1- (n)		(1)	(4)	(6)	
 SCL (mm)		161	319 ± 31	246 ± 53	
 Mass (g)		532	9,035 ± 2,269	4,108 ± 1,993	
Emb-D- (n)	(4)	(14)	(3)	(3)	
 SCL (mm)	138 ± 12	148 ± 23	229 ± 18	202 ± 11	
 Mass (g)	327 ± 79	452 ± 184	3,200 ± 625	2,245 ± 321	
Emb-C- (n)				(1)	
 SCL (mm)				171	
 Mass (g)				1,200	
Emb-B- (n)		(1)	(1)		
 SCL (mm)		153	195		
 Mass (g)		509	1,975		
Emb-A- (n)	(1)				
 SCL (mm)	139				
 Mass (g)	352				
On Land- (n)				(3)	
 SCL (mm)				284 ± 14	
 Mass (g)				5,800 ± 435	
Note:

Sample sizes shown in parentheses, means are followed by ± 1 standard deviation.

We report on turtle density and biomass (all species) as well as species-specific density and biomass for MP turtle and SN for the most productive wetlands in Table 4. Note that because of the small size of Goldfish pond, density, and biomass per hectare are artificially inflated.

Table 4 Densities and biomass of captured turtles at selected productive wetlands in the Tommy Thompson Park, Toronto, ON 2016–2017 data.

Selected wetland	Area (ha)	Density-individual per ha: all species (MP, SN)	Mass all species (kg)	Mass: MP, SN (kg)	Biomass: Total (MP, SN) (kg/ha)	
Goldfish pond	0.05	660 (560, 20)	22.939	9.988, 3.963	459 (200, 79)	
Triangle pond	0.8	69 (61, 6)	60.470	13.204, 45.505	76 (17, 57)	
Cell-1	6.0	2.8 (0.2, 2)	74.637	0.532, 68.487	12.4 (0.1, 11.4)	
Embayment D	7.0	4.7 (3.4, 0.9)	32.886	8.403, 21.642	4.7 (1.2, 3.1)	
Total	13.85	10 (7.4, 1.7)	190.932	32.127, 139.597	13.78 (2.3, 10)	
Note:

Entries represent the total for all species including Red-Eared Sliders and Blanding’s, with detail inside brackets for Painted (MP) and Snapping turtles (SN).

Our first method of quantifying movement through the site was based on the locations of recaptured turtles, and provided a minimum value of exploratory movement. Out of the 35 recaptured MP turtles, seven (20%) switched water bodies between captures, while only one (7%) recaptured SN was first captured in a different location.

We also captured 52 fish of 10 species as by-catch, including 37 common carp (C. carpio) (see Table 5). Of note, we had not captured common carp in 2016 in Embayment-D, but carp were able to migrate into the embayment in 2017 as a consequence of record high lake levels in spring. In general, we were able to release by-catch unharmed. However, there were a few occasions where fish caught in the mesh of the hoop traps had drowned, or had been mortally injured and/or partially consumed by trapped or nearby SN.

Table 5 Incidental fish capture (bycatch) by wetland area at the Tommy Thompson Park.

Number of fish (2016, 2017).

Common name	Species	Goldfish	Triangle	Cell-1	Emb-D	Emb-C	Total	
Common carp	Cyprinus carpio			8,13	0,14	1,1	9,28	
Pumpkinseed	Lepomis gibbosus			0,1		0,2	0,3	
Creek chub	Semotilus atromaculatus		0,2				0,2	
Northern pike	Esox lucius			0,2			0,2	
Brown bullhead	Ameiurus nebulosus			0,1		0,1	0,2	
Temperate bass	Morone chrysops			0,2			0,2	
Largemouth bass	Micropterus salmonides			0,1			0,1	
Gizzard shad	Dorosoma cepedianum			0,1			0,1	
Black crappie	Pomoxis nigromaculatus	0,1					0,1	
Bluegill	Lepomis macrochirus		0,1				0,1	
Total		0,1	0,3	8,19	0,14	1,4	9,43	

Habitat use by Snapping and Blanding’s turtles

Snapping turtles: We monitored a total of 21 SN (eight males and 13 females), and 14 of these turtles (three males and 11 females) were observed visiting more than one wetland. Of the 11 SN that were initially captured in a wetland and then explored other water bodies (eight females and three males), 10 returned to their initial capture point for overwintering. The mean minimum distance travelled (straight line distance) was 0.5 km (0–2.5 km) for males and 0.76 km (0–2.6 km) for females. One female that was captured on land (in July 2017) left TTP to overwinter at an island, 1.6 km away. We tracked the turtles to hibernacula from mid-September to end of October (Fig. 2). However, a few SN in Embayment-D relocated in early November 2016, likely due to sudden drop in the water level caused by the demolition of a beaver dam located inside the fish gate. All of the SN overwintered in shallow water (less than one m of water above the hibernaculum). Overwintering sites included: under vegetation mats (12), under logs, root balls and stumps (10), in mud flats with no vegetation above (four), at the edge or inside of beaver lodges (two), and under artificial structures (a wooden dock and a concrete slab) (see Table 6).

Figure 2 Map of overwintering locations of tracked Snapping and Blanding’s turtles at TTP 2016–2017.

Character codes: first letter represent species (B, Blanding’s; S, Snapping) and following numeral indicates the specific individual turtle.

Table 6 Movement of turtles equipped with a VHF transmitter during 2016/2017 at Tommy Thompson Park.

Species	Notch number	Sex	Initial capture point	Winter 2016	Winter 2017	Wetlands used over study period (some are used multiple times)	Minimum distance covered 2016–2017 (km)	Size estimation of habitat used (ha)	
Blanding’s	1	F	G	G	G	C1, C2, C3, EA, EB, EC, G, T	6.1	112	
Blanding’s	2	F	G	G	G	EA, EB, G	0.8	18	
Blanding’s	3	F	G	G	G	EA, EB, G	1.7	18	
Snapping	2	F	T	T	T	EB, T	0.4	3	
Snapping	5	F	C1	C1	C1	C1	0	6	
Snapping	8	F	C1	C1	C1	C1, C2	1.2	17	
Snapping	9	F	C1	C1	C1	C1, ED	0.7	30	
Snapping	11	F	ED	ED	ED	C1, ED	0.7	30	
Snapping	12	F	ED	ED	No signal	ED	0	8	
Snapping	13	F	ED	ED	ED	C1, ED	0.7	30	
Snapping	17	F	C1	C1	C1	C1, C2	0.4	17	
Snapping	22	F	T		T	EB, T	0.6	3	
Snapping	23	F	C1		C1	C1, ED	0.7	30	
Snapping	25	F	Road ED-C1		C1	C1, EC, ED	1.6		
Snapping	26	F	Road C1		ED	C1, ED	0.3	30	
Snapping	27	F	Road ED		Ward Isl.	EA, ED, WI	2.6	168	
Snapping	1	M	G	G	Not tracked	G	0	0.1	
Snapping	3	M	C1	T	Not tracked	C1, C2, C3, T	2.5	60	
Snapping	4	M	EB	EB	EB	EB	0	2	
Snapping	6	M	C1	C1	No signal	C1	0	6	
Snapping	7	M	C1	C1	Not tracked	C1, ED	0.7	30	
Snapping	10	M	ED	ED	No signal	ED	0	8	
Snapping	15	M	ED	ED	No signal	EC, ED	0.8	31	
Snapping	16	M	ED	ED	ED	ED	0	8	
Note:

Individual water bodies inside the wetland complex are identified with a two-letter code as such: C1, C2, C3 (Cell#); EA, EB, EC, ED (Embayment A, B, C, or D); G, Goldfish pond; T, Triangle pond.

Blanding’s turtles: We tracked three female Blanding’s turtles in 2016 and in 2017. All Blanding’s turtles explored at least three wetlands and all three returned to overwinter in the same small pond. The most active Blanding’s turtle traveled a minimum distance of 2.9 km over the summer of 2016 and 3.2 km in 2017. These turtles overwintered together in both study years, in an area measuring approximately 10 m2 located in the deepest section of Goldfish pond (>1.5 m deep). The Blanding’s turtles returned to their overwintering ponds by mid-September and remained active in that pond until mid-November of each year.

Visual surveys

Opportunistic basking surveys conducted by staff and citizen scientists (Table 7) identified the presence of two further species: a Northern Map turtle (G. geographica) in Cell-2 and a suspected Spiny Softshell turtle (Apalone cf. spinifera) in Cell-1 and in Cell-2.

Table 7 Years of detected turtle species by visual surveys at TTP.

2004–2017 data.

Species/wetland	Goldfish	Triangle	Cell-1	Cell-2	Emb-D	
Painted	2004–2017	2012–2017	2004–2012, 2014		2004–2017	
Snapping			2012			
Blanding’s	2007–2017					
Map				2015, 2016		
Spiny softshell			2006, 2012, 2015, 2016	2017		

Turtle nests were opportunistically discovered throughout the study period, either by observing a nesting turtle or by discovering a nest site that had been depredated. During our study period, we came across five SN actively laying eggs along the gravel shoulder of the main spine road or in gravel side roads and trails. We did not observe any MP turtles actively laying eggs in a nest. Nesting activity was first observed in mid-June and as late as early August. We also discovered a number of other nest sites by finding dug out holes with scattered empty shells at the surface, presumably depredated. We did not identify any of the depredated nests to species.

Results of the DNA analysis

Microsatellite locus Eb19 did not amplify, and loci GmuD55 and GmuD93 produced genotypes for the positive control that did not match those in the reference data from Davy, Bernardo & Murphy (2014). These three loci were excluded from further analyses. Neither turtle was strongly assigned to any previously profiled population: probability of assignment was <0.65 in all cases for the Bayesian analysis, and <0.45 in all cases for the frequency-based analysis. This result suggests that the two TTP Blanding’s turtles originated from a population that was not represented in the reference data (including samples from the proximate Golden Horseshoe population), and that these assignment results should only be used to infer genetic similarity, but not genetic origin. The Bayesian analysis assigned the first individual most strongly to the Kawartha Highlands, and the second most strongly to Parry Sound District (the eastern shore of Georgian Bay).

Discussion

Two turtle species, the MP and the SN, have successfully colonized our study site and established breeding populations. Both species are found in Lake Ontario at various sites within a few kilometres from TTP. Turtles were present in all surveyed areas of the TTP wetlands complex and were found in high densities in the two isolated ponds. Elsewhere, Painted turtle densities of greater than 100 individuals per hectare have been reported at a suburban site in North Carolina (Eskew, Price & Dorcas, 2010), in New York (Zweifel, 1989) and in Nebraska (Iverson, Baker & Dishong, 2006). The juveniles of MP and SN were discovered opportunistically, as well many nests and nesting SN, which we interpreted as a sign of well-established populations. We attribute the lack of observed nesting MP turtles to timing of oviposition in the late afternoons or early evenings (Rowe, Coval & Dugan, 2005), a time when researchers were not at TTP.

Blanding’s turtles have started to re-colonize TTP, with sightings recorded since 2007, but remain rare, and the lack of observed males suggests that this colonization remains precarious. The nearest known occurrence of Blanding’s turtles is approximately 15 km inland in the Rouge River. Blanding’s turtles can travel considerable distances. Females have been documented travelling distances of up to 3,479 m during nesting forays and nesting sites have been found as far as 1,850 m away from wetlands (Millar & Blouin-Demers, 2011), however, the genotype of the TTP samples could not be assigned to any known population, and thus provides no evidence as to how this species arrived at TTP. The single observation of Northern map turtle suggests that this species has only recently begun to colonize TTP. Map turtles generally require high-quality water that supports mollusc prey and are found in clean lakes and rivers. The TTP study site does support some crayfish species (both native and non-native) and snails, so we would expect to eventually find Map turtles in greater numbers. In urban landscapes, introduced exotic species common in the pet trade, such as the Red-Eared Slider turtle, are ubiquitous in easily accessible wetlands and can have detrimental effects upon native turtles (Cadi & Joly, 2004). Although we found some adult Red-Eared Slider turtles throughout the wetland complex, we did not find any evidence of juveniles, suggesting that perhaps this exotic population was not successfully nesting. The presence of the Spiny Softshell (Apalone cf. spinifera) at our site remains enigmatic. Although historically present but rare in Lake Ontario, the native Eastern Spiny Softshell (A. spinifera spinifera) is now considered locally extirpated in Lake Ontario (Committee on the Status of Endangered Wildlife in Canada (COSEWIC), 2016). Recent sightings of Softshell turtles in the lake have been attributed to non-native species, likely released.

MP and SN are sympatric (Bodie, Semlitsch & Renken, 2000), however, these species have shown some different habitat preferences. At TTP, MP turtles preferred smaller ponds with little or no fish and abundant shoreline vegetation consistent with past findings (Hughes, Tegeler & Meshaka, 2016; Marchand & Litvaitis, 2004), whereas SN were found in each wetland. Surprisingly, SN represented more than 20% of all individuals and 73% of the biomass of all turtles captured at our site. The MP to SN ratio of 3.5:1 was less skewed toward MP turtles when compared to an 11:1 ratio found from a previous study of multiple sites in Lake Ontario (DeCatanzaro & Chow-Fraser, 2010) and a ratio of 8.5:1 at the coastal Dunnville marshes near Lake Erie (Brown, 2016). Although the relative preponderance of SN at TTP compared to natural sites stands out as unusual, it is not unique in artificial pond systems. A study of artificial ponds in Pennsylvania found an overall 1.3:1 ratio between these species, with some ponds having a majority of SN (Hughes, Tegeler & Meshaka, 2016). A study of 88 wetlands, refuges and golf courses in upstate New York found a ratio of 1:1.5, with SN showing better body condition and more even sex ratios in golf course wetlands than in protected areas (Winchell & Gibbs, 2016). The TTP SN density of 1.7 individuals per ha (biomass 10 kg/ha) approaches that of other sites in Ontario ranging from 2 to 2.7 turtles per ha (biomass 13–18 kg/ha) at shallow artificial lakes, but nowhere near the 58 turtles per ha (biomass 313 kg/ha) at a nutrient enriched pond (Galbraith et al., 1988). Cell-1 had many SN but was mostly devoid of MP turtles. TRCA had documented several sightings of MP turtles in Cell-1 from 2004 to 2012, but none in 2013 and only one in 2014.

Common carp have destructive effects on shallow freshwater ecosystems including reducing aquatic macrophyte cover (Lundholm & Simser, 1999) due to the carp’s bottom-feeding behaviour (Fletcher, Morison & Hume, 1985; Koehn, 2004) and increasing water turbidity (Chow-Fraser, 1999). We suspect that the fish gate, equipped with seven cm grate width, in Cell-1 has allowed smaller carp (total length <50 cm) to access the cell in the spring (French, Wilcox & Nichols, 1999). These carp may have grown over the summer to larger than 50 cm total length thus becoming trapped by the fish gate inside the cell and unable to return to the lake. The presence of carp on its own does not explain the lack of MP turtles in Cell-1 given that MP turtles and carp coexist in Embayment D. However, the continuous presence of large carp in combination with the clay substrate in Cell-1 that is easily disturbed and suspended (Daviescolley et al., 1992; Harter & Mitsch, 2003) could have caused the degradation of the emergent vegetation. Turbidity, per se, does not affect the ability of Painted turtles to locate prey species (Grosse, Sterrett & Maerz, 2010) but turbidity does negatively affect vegetation growth necessary for omnivorous species such as the MP turtle (Parkos, Santucci & Wahl, 2003).

Although SN are also omnivorous and prefer less turbid waters (Bodie, Semlitsch & Renken, 2000), adults may be better equipped to co-exist in carp-modified environments (Paterson, Steinberg & Litzgus, 2012). SN will readily feed on fish and evidence from our by-catch suggests that fish are abundant in Cell-1.

Safe and accessible terrestrial habitat between wetlands is an important characteristic of successful wetland complexes (Burke & Gibbons, 1995; Gibbons, 2003). At our site, turtles could move freely through the park, as there was very little vehicular traffic, and they could also use the terrestrial habitat to gain access to a variety of suitable nesting sites. The safety of the terrestrial habitat is a likely contributor to the evenness of the sex ratios and the presence of turtles in all surveyed wetlands. Given the amount of movement we documented between water bodies, we are inclined to consider the population sex ratios at the wetland complex level. The sex ratios of both MP and SN across the whole wetland complex were remarkably even. The evenness of the sex ratio at our site is notable given that turtle sex ratios have been showing a male bias in North America (Gibbs & Steen, 2005) as well as in Ontario (DeCatanzaro & Chow-Fraser, 2010) attributed to the higher risks of vehicle collisions to females searching for nesting sites by the roadside (Aresco, 2005; Steen et al., 2006). Possible sources of observed sex-ratio bias could stem for local climatic conditions (both MP and SN exhibit temperature-dependent sex determination), differential exposure to predators (Marchand & Litvaitis, 2004) and capture methods (Ream & Ream, 1966). Dupuis-Desormeaux et al. (2017) found a heavily male-skewed sex ratio in MP turtles at a nearby urban wetland site (Heart Lake Rd.) with a history of turtle road mortality. The TTP and Heart Lake Rd. study sites share similar general climatic conditions (microhabitat differences aside), similar predator presence and used the same capture methodology. Two key differences stand out between the TTP and Heart Lake Rd. study sites: differences in traffic volume (fewer than 50 vehicles per day at TTP, mostly dump trucks, vs. 5,000–7,500 vehicles per day at Heart Lake Rd.) and differences in the timing of that traffic (7:30 am to 4:15 pm, weekdays only at TTP, vs. 24 h per day, 7 days per week at Heart Lake Rd.). Therefore the times of highest traffic volume at the TTP site avoided the peak nesting times (early morning or late afternoon) for these two turtles species (Obbard & Brooks, 1981; Rowe, Coval & Dugan, 2005). We did not set out to study road mortality vs. sex-ratios at TTP, and consequently have no data on TTP road mortality; however, given the above described similarities and differences between sites, we are inclined to speculate that the lack of vehicular traffic at TTP contributes to the evenness of the observed sex-ratio.

The results of our recapture and our tracking study revealed that MP, SN, and Blanding’s turtle species visited all of the surveyed water bodies inside the wetland complex. Furthermore, some Snapping and Blanding’s turtles explored large portions of the park during the summer yet displayed site fidelity toward their ponds of capture for overwintering. Nesting site fidelity has been documented in both Blanding’s (Standing, Herman & Morrison, 1999) and SN (Loncke & Obbard, 1977) as well as overwintering site fidelity (Brown & Brooks, 1994; Innes, Babbitt & Kanter, 2008; Meeks & Ultsch, 1990). SN hibernacula were distributed throughout the wetland complex and we noted three pairs that overwintered in close proximity to each other (two male–female pairs and one female–female pair). However, we did not find evidence of mass congregation at a communal site as has been reported in some other populations (Meeks & Ultsch, 1990). The structure of the hibernacula was similar to what has been reported in past studies at natural wetland sites (Brown & Brooks, 1994; Paisley et al., 2009) and in that respect, the overwintering sites at the TTP artificial wetland site mimicked that found in natural ponds.

The overland movement of turtles will increase their exposure to predators. At TTP, we have documented many potential turtle predators, including Eastern coyote (Canis latrans), raccoon (Procyon lotor), otter (Lontra canadensis), and mink (Neovison vison). We noted several MP turtles with shell markings consistent with predator encounters, including bite marks, puncture wounds, missing carapace pieces, scratch marks, and missing limbs. We also discovered many depredated turtle nests. Turtles seem to be colonizing the created wetlands despite the presence of predators, but we would have to monitor survivorship over time to understand exactly how predation impacts population growth at TTP.

Conclusion

The two Blanding’s turtles were not strongly assigned to any previously genotyped populations (Davy, Bernardo & Murphy, 2014) based on their microsatellite genotypes, indicating that they belonged either to a remnant population of turtles from the TTP area, or that they had been introduced to TTP from an entirely different area. Of the populations previously profiled by Davy, Bernardo & Murphy (2014), the genetic profiles of the TTP Blanding’s turtles were most similar to the closest candidate populations, which were in the Kawartha Highlands and the east shore of Georgian Bay. Thus the genetic data suggest that the TTP Blanding’s turtles probably belong to a remnant local population, although the possibility that they were introduced from elsewhere cannot be completely excluded. This result is most consistent with a colonization of TTP by Blanding’s turtle from nearby areas (possibly the Rouge River), and a pattern of genetic isolation by distance between Blanding’s turtle populations on the Canadian Shield, and remnant populations near Lake Ontario. Confirming this hypothesis would require genotyping a robust sample of turtles from the Rouge or elsewhere along the north-western shore of Lake Ontario, as this would allow frequency-based analyses of the similarity between the Rouge and TTP individuals. However, collecting sufficient samples from the Rouge would be challenging as the original population apparently declined rapidly following the urbanization of the Greater Toronto Area.

We conclude that the artificial wetland complex at TTP has provided adequate habitat to attract four native species and that two of these species are nesting and recruiting juveniles to the TTP populations. Many other factors can influence colonization success, including the connectivity of the surrounding landscape matrix, specific habitat suitability, predator density, and the particular dispersal ecology of neighboring seed populations. We suggest regularly assessing the progress in the colonization of each individual water body of this artificial wetland complex and of extending the movement study to include specimens from the nearby reference wetlands to better assess colonization vectors.

Supplemental Information

Supplemental Information 1 Trapping and tracking data.

Click here for additional data file.

We would like to acknowledge the help of Peter Shuttleworth and Ben Shearer with trapping, processing and fish identification. We also would like to thank the many volunteer naturalists who monitor the wildlife at TTP on a daily basis, special thanks to Ian Sturdee, Don Johnston, and Paul Xamin. Both the Toronto Zoo Adopt-A-Pond Program and the Ontario Turtle Conservation Centre (Dr. Sue Carstairs) provided advice and hands-on training.

Additional Information and Declarations

Competing Interests

Author Contributions

Animal Ethics

Field Study Permissions

Data Availability

Emma Followes, Andrea Chreston, and Andrew Ramesbottom were employees of Toronto and Region conservation Authority at the time of the study.

Marc Dupuis-Desormeaux conceived and designed the experiments, performed the experiments, analyzed the data, contributed reagents/materials/analysis tools, prepared figures and/or tables, authored or reviewed drafts of the paper, approved the final draft.

Christina Davy performed the experiments, analyzed the data, contributed reagents/materials/analysis tools, authored or reviewed drafts of the paper.

Amy Lathrop performed the experiments, contributed reagents/materials/analysis tools.

Emma Followes conceived and designed the experiments, performed the experiments, contributed reagents/materials/analysis tools, prepared figures and/or tables, authored or reviewed drafts of the paper.

Andrew Ramesbottom analyzed the data, contributed reagents/materials/analysis tools, prepared figures and/or tables, authored or reviewed drafts of the paper.

Andrea Chreston contributed reagents/materials/analysis tools, prepared figures and/or tables, authored or reviewed drafts of the paper.

Suzanne E. MacDonald conceived and designed the experiments, performed the experiments, analyzed the data, authored or reviewed drafts of the paper, approved the final draft.

The following information was supplied relating to ethical approvals (i.e., approving body and any reference numbers):

This study was conducted under the supervision of York University’s Animal Care Committee (YUACC#2016-16W- and #2017-16W-R1).

The following information was supplied relating to field study approvals (i.e., approving body and any reference numbers):

Field experiments were approved Ontario Ministry of Natural Resources (Wildlife Scientific Collector’s Authorization numbers #1083601 and 1085922) and Endangered Species Act Permit for Species Protection or Recovery (AU-B-007-16 and AU-B-006-17).

The following information was supplied regarding data availability:

The raw data are provided as a Supplemental File.

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
