# Peer review of "Colonization and usage of an artificial urban wetland complex by freshwater turtles"

_PeerJ, doi:10.7717/peerj.5423_

## Round 0.1 · original submission · Major Revisions

I have now received comments and suggestion from two reviewers regarding your paper. Both recognize your extensive dataset and its importance. However, reviewer #1 has the opinion, which I agree, that you need to "dig" valuable information that it is beneath you rich data. In this regard, especially reviewer #1, had made several good suggestions that I strongly recommend that you fallow in order to improve the discussion of your manuscript. Reviewer #2 was more positive towards your manuscript, but also made nice suggestions that you should fallow. Based on these comments I think your paper needs MAJOR REVISIONS before acceptance. Please note that I am asking not for "cosmetic" changes, but instead that you take very seriously the reviewer recommendations. I wish you good luck with your revision and I am looking forward for your resubmission.

Reviewer 1 ·

Basic reporting

This is a very descriptive survey of turtles found in Tommy Thompson Park, a large created urban wetland complex in Toronto. It does not have any experimental design, no hypotheses to be tested, and no theories to explain the observations. The data provided are useful for further studies/investigations or for others to use in meta-analyses. There is no figure other than the maps showing location of the wetlands and where turtles overwintered. At least some of the information buried in tables should be made into figures. GIS maps to show the distribution of individual and population home ranges for the Painted and Snapping turtles would have been informative. The way in which they reported mean and SE needs to be streamlined (e.g. line 185)--perhaps journal has a required style. Figure 2 needs to be revised so that all of the labels do not obscure the information. Perhaps try to make inserts with an expanded view of those ponds and have lines pointing to the location of the overwintering sites.

Experimental design

There is no experimental design. The goal of this project is mainly to describe where and how many turtle species and individuals used the created wetland complexes. The CMR approach used in both years to determine population size of the two dominant species, as well as the tracking program they carried out were all appropriate and the data from these were very useful and credible. Nevertheless, they were not used to their fullest extent, and no spatial analysis was performed to determine how the turtles used their habitats. I also disagree with their analyses on a pond-by-pond basis rather than treating the population as a whole for the entire complex.

Validity of the findings

I have no concerns with the validity of what they have presented. Unfortunately, the authors did not ask more from their data. In particular, the movements and home ranges of the turtles would have added greatly to this study. I had many questions while reading their manuscript. Why did turtles only use some of the sites, and not others. What are the predators and how were they surveyed? Where is the reference wetland that was mentioned in the Discussion (line 346). How do these turtles behave relative to Snapping turtle populations in non-urbanized settings--provincial parks and other protected areas?

Additional comments

There is a lot of great information in this study that should be communicated to the conservation audience. For instance, the fact that Blanding's turtles can exist there at all is very surprising and interesting. The relatively large population size of the two dominant species in such a setting is similarly important to note. That said, there is missing information that could make this manuscript more valuable to scientists because as it reads currently, it is a technical report that would form the basis of a published paper. Missing things include home-range analyses, possible vehicular traffic estimates, location of turtles relative to the type of habitat features within the ponds, embayments, etc., presenting movement and use information according to the behavioural seasons (pre-nesting, nesting and post-nesting/overwintering), etc.

Reviewer 2 ·

Basic reporting

In general, the article is quite interesting, with a well-defined objective. The authors contextualized the question satisfactorily, and the practical implications is very clear.

- Background/context

Line 47-51: The territorial proportion and conservation status of the wetlands are relevant information, but insufficient to highlight their importance. The authors could discuss more about the ecosystem services that the wetlands provide.


- Structure, figs, tables

Figure 1: I believe the line between the image and the "Embayment B" label has been misconfigured. For a standardization criterion, I suggest to include a line between the image and the "Embayment D" label.

Figure 2: Too much information in the figure, which makes it confusing for the reader. It does not seem to facilitate the understanding the information that was presented in the text.

Table 2: Standardized the abbreviations in table and title.

Table 4: I suggest that the authors report the density and biomass for all the wetlands. It is only three more!

Table 5: The table title is uninformative. Formatting (font type and font size) need to be standardized.

Table 6: What is “na”? – Make an observation or explain in the title.

Experimental design

- Some points of methods need a more detailed description.

Line 84-90: A more detailed characterization of water bodies is necessary. Do they differ only in size and connectivity with Lake Ontario?

Line 97-99: The authors could provide some more relevant information, such as mesh size and size of the traps.

Line 110-112: Why basking traps and seine net were trapped only in 2016? Could this have affected the success of the recaptures in 2017?

Line 113 -114: Need a more detailed explanation of CRM protocol, or at least a reference for consult.

Line 119-121: Why some wetlands were sampled only in one year? Does this have implications for population estimates?

Line 127: A character (multiplication) is missing in the formula.

Line 145-147: It was not clear why new radios were added to females, while others radios was removed from males.

Validity of the findings

No comment

Additional comments

The modification of aquatic and semi-aquatic environments because of urbanization is a global problem that affects different populations of a wide variety of organisms that depend in some way on these environments. In the specific case of turtles, which have certain intrinsic biological characteristics, such as low recruitment rates in their populations, low rates of body growth and late sexual maturity, these modifications can be critical. On the other hand, some turtle species have the ability to adapt to non-optimal conditions and therefore to establish themselves with some success in environments where natural conditions have changed. Thus, the study presents a very relevant issue from the point of view of conservation and restoration of both wetlands and native turtle populations in the Lake Ontario region.

Specific comments:

Line 168-169: Why differentiate between new individuals and recaptures in the year of 2016? This information is not used for the estimates of population size, and the study is limited to data from 2016/2017. By highlighting the recaptures in 2016 the authors can confusing the reader about how the population estimates were calculated.

Line 175-176: The definition of “species assemblage” is not related to the proportion between different species as presented in the manuscript. I suggest to the authors to see Fauth et al. (Am. Nat., 1996, V. 147 pp. 282-286) and Stroud et al. (Ecology and Evolution, 2015, V. 5(21), pp.
4757-4765) for a discussion about the terminology.

- It is unnecessary to present both proportions! I suggest keeping only the most direct one (second proportion).

Line 185-188: Data presentation need to be standardized. For one species size data is presented with standard deviation and for another it is not. In addition, size data are presented with minimum and maximum, while mass data is not.

Line 189-191: Replace "weight" with "mass". Standardize the unit presentation.

Line 220-221: Consider substitute “visiting other wetlands” for “visiting more than one wetland".

Line 232-233: Consider substitute “explored at least two other wetlands” for ““explored at least three wetlands”.

Line 288-291: The authors use the sentence “species assemblage ratio”. See the suggestions for the lines 175-176.

-The ratio between species is always skewed towards Painted turtles, therefore the sentence “The TTP Painted to Snapping turtle species assemblage ratio of 3.5:1 was skewed towards Snapping turtle compare to…” is incorrect. Although the intention was compare this result to the ratios found in other areas, the only difference between areas is the force of skew and not the direction of skew. I suggest the authors rewriting the phrase.

Line 292-294: In the text is not clear if the ratio presented is between Painted turtle and Snapping turtle. I suggest substitute “… found an overall 1.3:1 ratio” for “… found an overall 1.3:1 ratio between this species”.

---

## Round 0.2 · Minor Revisions

I sent your papers for one of the original reviewers. Reviewer 1 did not feel the need to re-review, but Reviewer 2 recommends further "minor revisions". Please, make sure that you take care of them all, and resubmit your manuscript again. We are almost there!

Reviewer 2 ·

Basic reporting

no comment

Experimental design

- Original primary research within Scope of the journal.

Yes

- Research question well defined, relevant & meaningful. It is stated how the research fills an identified knowledge gap.

The paper brings an interesting study with relevant implications for conservation and ecological restoration. However, the new objective presented is less clear and concise than the previous one. The distinction and the hierarchical relationship between the main objective and the specific objectives should be clearer. For example, in my opinion, investigating the colonization of a complex of artificial wetlands by turtle species is the main objective, while determining the populations of origin of Blanding's turtles, and the distribution of species along the wetlands complex are specific objectives.

- Rigorous investigation performed to a high technical & ethical standard.

Yes, all the technical and ethical standards were performed.

- Methods described with sufficient detail & information to replicate.

Some details of the sampling design still need to be better explained in the manuscript. Although the authors have justified some choices in the response letter, without these justifications readers may be confused (For more details see specific comments section).

Validity of the findings

no comment

Additional comments

In general, the authors answered almost all my questions in a satisfactory manner. Although I lacked a more standardized sampling design, especially with regards to the sampling of different wetlands and different years, this lack did not affect the results and conclusions presented in the manuscript. Anyway, I suggest that the authors include in the manuscript some of the answers presented in the response letter that have not yet been added, as this will facilitate readers comprehension and help to resolve doubts that may arise during the reading. I point out the answers that I believe should be included in the manuscript in the specific comments section.

Specific Comments

- Answers that should be included in the manuscript:

Line 110-112: Why basking traps and seine net were trapped only in 2016? Could this have affected the success of the recaptures in 2017?
The seine net was only used one day in 2016 (one snapping turtle captured) and was unavailable in 2017 and the basking traps had little success in 2016 and were not used in 2017. Unlikely, given how little success they had in 2016.

Line 119-121: Why some wetlands were sampled only in one year? Does this have implications for population estimates?
Wetlands that had very few turtles captured in 2016 were not re-sampled the following year.

Line 145-147: It was not clear why new radios were added to females, while other radios was removed from males.
In 2016 we tried to get an even number of males and females. After an initial data investigation, we noticed that the females were moving more than the males. In 2017 we were more interested in the movement of females, so added transmitters only to females. When we recaptured males in 2017, we removed the old VHF transmitters (most had stopped working anyway). We added a line of text to better describe our motivation.

- Table 4: Replace "weight" with "mass".

- The presentation of the results and discussion about the genetic data of Blanding's turtle is confusing. In the passage "Microsatellite genotypes for the two Blanding's turtles were not strongly assigned to any previously genotyped populations (Davy et al., 2014) but were most strongly assigned to the previously profiled populations in the Kawartha Highlands and the east shore of Georgian Bay.", it is unclear whether the two individuals sampled in the TTP are associated with any population cited in Davy et al. 2014. Do the populations of Kawartha Highlands and the east shore of Georgian Bay refer to other paper?

---

## Round 0.3 · accepted · Accept

The authors have done a fantastic job by reviewing this manuscript. I am glad to accept in its present form. Congratulations! Well done!

#